# An Origami Heat Radiation Fin for Use in a Stretchable Thermoelectric Generator

**DOI:** 10.3390/mi11030263

**Published:** 2020-03-04

**Authors:** Momoe Akuto, Eiji Iwase

**Affiliations:** Department of Applied Mechanics, Waseda University, 3-4-1 Okubo, Shinjuku, Tokyo 169-8555, Japan; akuto@iwaselab.amech.waseda.ac.jp

**Keywords:** energy harvesting, thermoelectric generator, origami, stretchable device, flexible device

## Abstract

Recently, some studies have addressed the use of a folded substrate to realize stretchable electronic devices including stretchable thermoelectric generators (TEGs). However, the utilization of the folded substrate as a heat radiation fin has not been achieved. Herein, we have proposed the construction of a TEG with an origami-like folded structure substrate called an “origami-fin” that can achieve a high heat radiation performance and is also highly stretchable. The origami-fin increases the stretchability of the TEG by bending a non-stretchable material into a folded shape, and it also works as a heat radiator because of its large surface area compared to that of a flat structure. We evaluated the heat radiation performance of the origami-fin and the stability of the performance when it was stretched. The results demonstrate that the origami-fin works as a heat radiator and enhances the output of the TEG, while also exhibiting a high stretchability with only a slight output reduction.

## 1. Introduction

Thermoelectric generators (TEGs) are devices used to convert thermal energy into electrical energy using the Seebeck effect by providing a temperature difference to thermoelectric (TE) elements [1,2]. Recently, the distribution of many sensors in various places in the environment has been proposed, and TEGs represent a potential power source for those sensors [3,4,5,6]. For example, there have been reports of bendable TEGs [7,8,9,10,11,12,13,14,15] and stretchable TEGs [16,17] for IoT devices or wearable devices that attach to curved heat sources (such as piping, car bodies, and the human body). In particular, a stretchable TEG is needed so that it can be attached with low thermal contact resistance to curved heat sources and harvest energy from them [16]. Furthermore, the TEG’s heat radiation performance is also important for power generation. Generally, the heat radiation performance is improved by attaching a heat radiation fin to the TEG. It is, however, difficult to attach a conventional heat radiation fin while maintaining stretchability because conventional fins are made from rigid and non-stretchable materials such as metals or ceramics. Thus, a stretchable TEG integrated with a fin for heat radiation has been necessitated. In previous studies, although the production of stretchable TEGs that used a folded substrate was reported, the folded substrate was not considered to be a heat radiation fin [17]. Therefore, we decided to develop a stretchable TEG with a high heat radiation performance that uses a folded substrate as a heat radiation fin.

In this paper, we fabricated TEGs with origami-fins and measured their output voltage. In addition, the output voltage of the TEGs was simulated using a finite element method (FEM) model. Based on the output voltage obtained from both the measurements and the simulation, and the heat transfer coefficient between the origami-fin and air that was calculated from those results, the heat radiation performance of the origami-fin and the stability of the TEG during stretching were evaluated.

## 2. Design and Fabrication

Figure 1 shows the TEG with the origami-fin which we proposed. The stretchability of the TEG was produced by placing an origami-fin, which is a folded substrate, between the TE elements. In addition, the origami-fin acted as a heat radiation fin because its folding increased the surface area of the substrate. The local bending of the substrate within the origami-fin made the whole device stretchable, so the substrate material was not required to have inherent stretchability. Therefore, we were able to achieve a high heat radiation performance by using a substrate material with a high thermal conductivity. We used non-stretchable copper-polyimide film as the substrate material because it has a higher thermal conductivity than stretchable elastomeric substrate materials. Furthermore, since the origami-fin contains an undeformable flat plate region, it was possible to use rigid TE elements that exhibited a high thermoelectric conversion performance. In this study, a BiTe-based element, which is known to have a high figure of merit (*ZT*) of approximately 1, was selected.

Figure 2 shows the fabrication process of the TEG with the origami-fin. We first fabricated the upper and lower substrates using copper-polyimide films (polyimide: 25 μm-thick, copper: 8 μm-thick). Using a UV laser processor (OPI Corporation, OLMUV-335-5A-K, Saitama, Japan), the copper layer of the film was scratched and patterned. Then, the film was cut by a UV laser processer. After the substrates were folded manually, a single pair of TE elements (TOSHIMA Manufacturing, Saitama, Japan, p-type: Bi_0.3_Sb_1.7_Te_3_, n-type: Bi_2_Te_3_, 1.5 mm × 1.5 mm × 1 mm) and wirings were soldered onto the substrates. The TE elements were assembled in a π-type structure, which is a typical structure of the out-of-plane TEGs, electrically in series and thermally in parallel [2].

The fabricated TEGs are shown in Figure 3. As shown in the top of Figure 3, we defined an evaluation unit to contain one flat plate region and one folded region and defined the width as *w* (mm), the substrate surface area as *S* (mm^2^), and the substrate projected area as *P* (mm^2^). Each TEG was fixed to an aluminum plate (25 mm × 25 mm × 0.5 mm) with an adhesive that had high thermal conductivity (Cemedine, SX1008, Tokyo, Japan) to maintain its shape and facilitate repeated experiments. First, in order to evaluate the heat radiation performance of the origami-fin, five types of TEGs containing origami-fins with various surface areas (*S* of 18 mm^2^, 27 mm^2^, 54 mm^2^, 99 mm^2^, and 144 mm^2^), were fabricated (Figure 3i–v). All of these devices had a width of 4 mm, which means that all the projected areas were equal. The TEG with *S* of 18 mm^2^ was configured in a flat state with the same projected and surface area. The other four TEGs had larger surface areas that ranged from 1.5 to 8 times the projected area. Next, in order to evaluate the stability of the output when the origami-fins were stretched and contracted, three types of TEGs with origami-fins that had the same surface area and different widths were fabricated. Those TEGs had widths of 3 mm, 4 mm, and 6 mm, respectively, and were fabricated from a substrate with *S* of 27 mm^2^ (Figure 3ii,vi,vii). The TEG with *w* of 6 mm was maximally stretched so that it became flat, and the other two TEGs were contracted by folding the origami-fin.

## 3. Evaluation and Discussion

We obtained the open-circuit voltage (*V*_TEG_) (V) of the TEGs by measurement and simulation to evaluate the heat radiation performance of the origami-fin and the dependence of the output stability on stretching. We first measured the current (*I*) and voltage (*V*) characteristics of the fabricated TEGs using the measurement setup shown in Figure 4a and then calculated the *V*_TEG_ of the TEGs from the measured *I–V* characteristics. The TEG with an origami-fin that was fixed to an aluminum plate was placed on a heater using grease that had high thermal conductivity (Shin-Etsu Silicones, G-747, Tokyo, Japan). The temperature of the heater was controlled by a Peltier temperature controller (VICS, VTH1.8K-70S, Tokyo, Japan) and was increased from 313 K to 353 K in increments of 10 K. The room temperature was approximately 298 K while these measurements were made. The output current was measured by sweeping the applied voltage to the device using a source measurement unit (Keithley Instruments, 2614B, Cleveland, OH, USA). Three experiments were performed using each device. The measured *I–V* characteristics of the TEG with *w* of 4 mm and *S* of 27 mm^2^ are shown in Figure 4b as an example. We obtained the *V*_TEG_ from the *x*-intercept of the regression line calculated by the least squares method. Measurements were taken at different temperatures to examine the dependence of the TEGs on temperature, but we set the *V*_TEG_ at 323 K for evaluation.

To confirm the validity of the measurements, we simulated the *V*_TEG_ using an FEM. As shown in Figure 5, we created 2D simulations that were the same size and constructed of the same materials as the TEGs we fabricated. We set the temperature of the bottom of the device at 323 K and the heat transfer coefficient at 20 W/m^2^·K for the outer boundaries of the model. This value was set within an appropriate range for the value of the heat transfer coefficient between the solid material and natural convection air [18]. The thermal contact resistances between each material were not included in the model. We used this model to simulate a steady-state temperature distribution and calculated the *V*_TEG_ by multiplying the temperature difference between the top and bottom of the BiTe-based TE element (Δ*T)* by the Seebeck coefficient of the TE element. The Seebeck coefficient of the BiTe-based TE element at a temperature of 323 K was obtained from *S*_p_−*S*_n_ = 317.2 × 10^−6^ V/K by the material data sheet, where *S*_p_ and *S*_n_ are the Seebeck coefficients of the p-type and n-type elements, respectively.

In addition, we introduced a heat transfer coefficient between a fin and the surrounding air (*h*_fin_) using the previously obtained *V*_TEG_ in order to understand the heat radiation performance of the origami-fin. To derive *h*_fin_, we used a simple model with only a TE element as shown in Figure 6a. The temperature of the bottom of the element was *T*_h_ (K), the temperature of the top of the element was *T*_c_ (K), and the temperature of the surrounding air was *T*_air_ (K). The thermal conductivity of the element was *κ* (W/m·K) and the thickness of the element was *t* (m). Assuming that heat flux is constant, if the heat transfer coefficient between the element and air is *h*_0_ (W/m^2^·K), the temperature difference Δ*T*_0_ (= *T*_h_−*T*_c_) occurring in the TE element can be obtained as follows:(1)ΔT0=Th−Tc=tκtκ+1h0(Th−Tair)

Alternatively, when a fin was attached to the top of the element, we assumed the heat transfer coefficient between the fin and air was *h*_fin_ and the temperature difference applied to the element was Δ*T*_fin_. Then, Δ*T*_fin_ can be obtained in the same way as Δ*T*_0_:(2)ΔTfin=Th−Tc′=tκtκ+1hfin(Th−Tair)
where *T*_c_’ is the temperature of the top of the TE element with the fin. Therefore, the relationship between Δ*T*_fin_ and Δ*T*_0_ can be described as follows:(3)ΔTfinΔT0=hfin(κt+h0)h0(κt+hfin)

Note that the thermal conductivity of the BiTe-based TE element, *κ*, is about 1 W/m·K. Thus, if the thickness of the element *t* is about 1 mm, as it was in this study, the order of magnitude of *κ*/*t* is 10^3^ W/m^2^·K. In contrast, the heat transfer coefficient between the solid materials and air was reported to be in the order of 10^0^ to 10^1^ W/m^2^·K during natural convection [18], which is very small compared to *κ*/*t* and can therefore be ignored in the calculations. So, Equation (1) can be expressed as:(4)$$\frac{\Delta T_{\mathrm{fin}}}{\Delta T_{0}}\approx \frac{h_{\mathrm{fin}}}{h_{0}}$$

If the *V*_TEG_ of the flat TEG with the smallest surface area among the TEGs we simulated (*S* of 18 mm^2^) is *V*_0_ and the *V*_TEG_s of the other origami-fin devices are *V*_fin_, the ratio of *V*_fin_ to *V*_0_ can be described as follows:(5)$$\frac{V_{\mathrm{fin}}}{V_{0}}=\frac{\Delta T_{\mathrm{fin}}}{\Delta T_{0}}$$

This is possible because the *V*_TEG_ (V) is equal to the product of the temperature difference *ΔT* (K) and the Seebeck coefficient of the element (*S*_p_−*S*_n_) (V/K). Therefore, from Equations (2) and (3), we can express *h*_fin_ using both the measured *V*_0_ and *V*_fin_ and an arbitrarily set *h*_0_ as follows:(6)$$h_{\mathrm{fin}}=\frac{V_{\mathrm{fin}}}{V_{0}}h_{0}$$

In this study, we set *h*_0_ at 20 and calculated *h*_fin_. The results with both measured and simulated *V*_TEG_ are shown in Figure 7.

As shown in Figure 7a, both the measured and simulated *V*_TEG_ were improved by the origami-fin. This indicates that the origami-fin worked as a heat radiation fin. In the simulation, the *V*_TEG_ increased as *S* increased. The measurement results showed that all TEGs with the origami-fin that had surface areas greater than 54 mm^2^ showed a higher *V*_TEG_ than the flat TEG with a surface area of 18 mm^2^. The *V*_TEG_ of the TEG with *S* of 144 mm^2^ was 1.9 times as high as that of the flat TEG. In other words, the heat radiation fin with an *h*_fin_ that was 1.9 times as large as that of the unfolded flat substrate was achieved by folding the substrate to increase its surface area. The value of the *V*_TEG_ changed when the type of TE element changed, but *h*_fin_ can be used as an index of the heat radiation performance of the origami-fin that does not change unless the setting of *h*_0_ is changed. These results confirmed that the output of the TEG can be increased by the presence of an origami-fin because the origami-fin can function as a heat radiation fin. Next, regarding the relationship between width and *V*_TEG_ and *h*_fin_ (Figure 7b), the measured and the simulated values were almost constant regardless of the device width. In the measurements, the *V*_TEG_ of the TEG with *w* of 3 mm was 92% of the *V*_TEG_ of the flat TEG with *w* of 6 mm. This means that even if the origami-fin width was reduced by 50%, *h*_fin_ would only decrease by 8%. These results confirmed that the stability of the origami-fin was not affected by deformation caused by stretching and contracting. This means that it was possible to use an origami-fin to construct a stretchable TEG integrated with a heat radiation fin.

## 4. Conclusions

We developed a TEG with both a high heat radiation performance and high stretchability by using a folded structure substrate called an origami-fin. We fabricated TEGs with origami-fins and evaluated them by measuring their output voltage and index of heat transfer coefficient. We first evaluated the heat radiation performance of the origami-fin using TEGs with various surface areas but the same projected area. As a result, the output voltages of TEGs containing origami-fins that made their surface areas larger than three times their projected areas were higher than the TEG that had the same surface area and projected area (the flat TEG). We obtained a heat transfer coefficient that was 1.9 times higher than that of the flat substrate which had a surface area of 18 mm^2^ using an origami-fin with a surface area that was eight times larger than that of the flat TEG. Next, we evaluated the performance stability of the origami-fin when subjected to stretching deformation. The results demonstrated that even if the TEG with the origami-fin was contracted to a width of 50% of its flat state by folding, the output voltage and heat transfer coefficient were still 92% of those in the flat state. These results indicate that the origami-fin works as a heat radiator to enhance the output while also exhibiting high stretchability with only a slight output reduction. Therefore, the TEG with the origami-fin could be useful for harvesting energy from curved heat sources.

## Figures and Tables

**Figure 1 micromachines-11-00263-f001:**
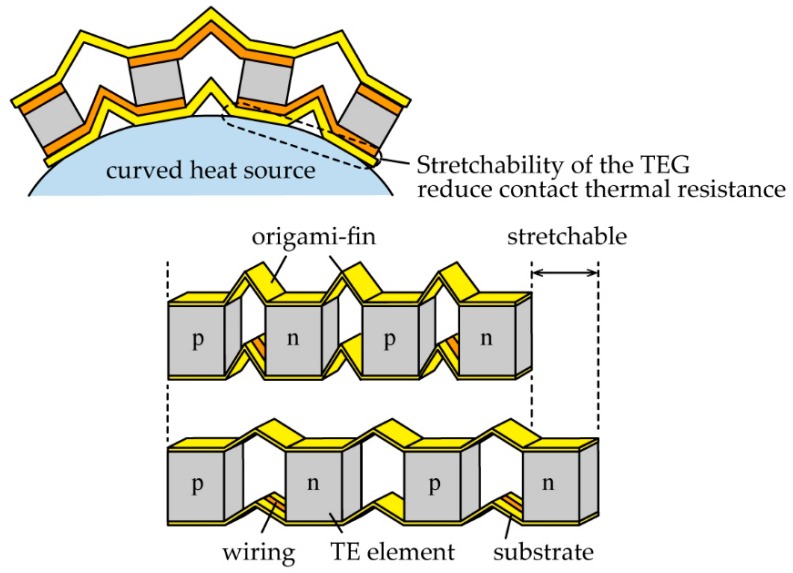
Schematic images of the thermoelectric generator (TEG) with the origami-fin. The origami-fin between the thermoelectric (TE) elements showed a high heat radiation performance and stretchability.

**Figure 2 micromachines-11-00263-f002:**
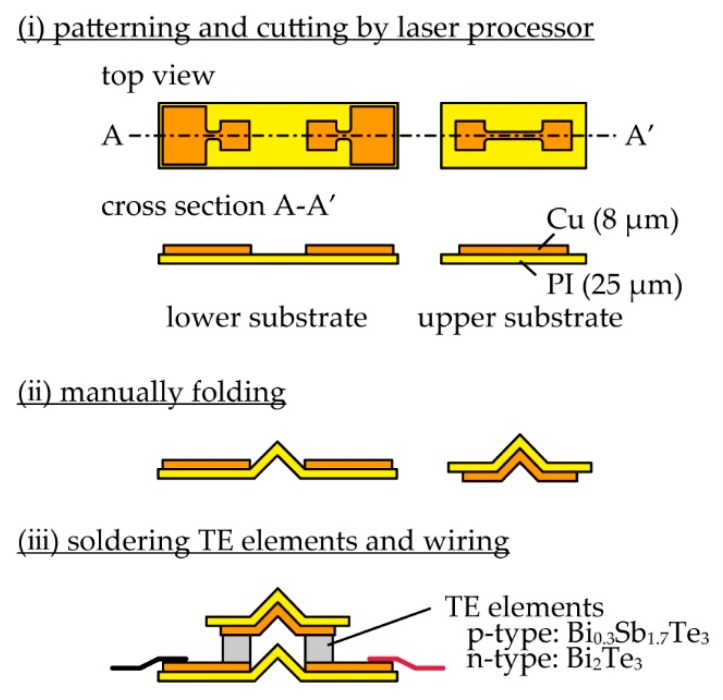
Fabrication process of TEGs with the origami-fin.

**Figure 3 micromachines-11-00263-f003:**
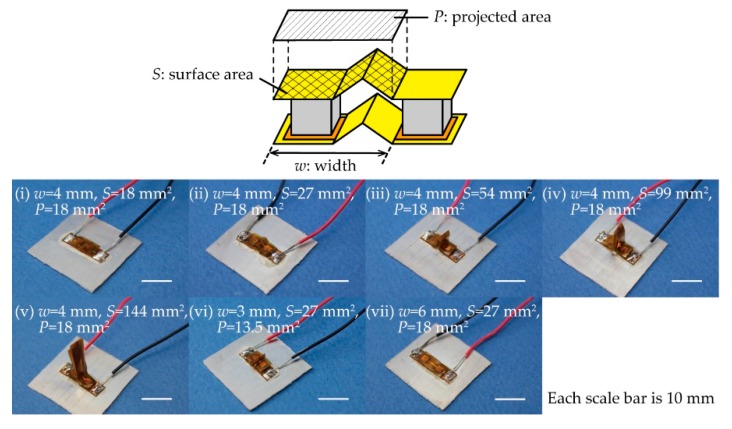
Definition of an evaluation unit and photographs of fabricated TEGs. TEGs with a width (*w*) of 4 mm and surface areas (*S*) of (**i**) 18 mm^2^, (**ii**) 27 mm^2^, (**iii**) 54 mm^2^, (**iv**) 99 mm^2^, and (**v**) 144 mm^2^. TEGs with widths of (**vi**) 3 mm and (**vii**) 6 mm and a surface area of 27 mm^2^.

**Figure 4 micromachines-11-00263-f004:**
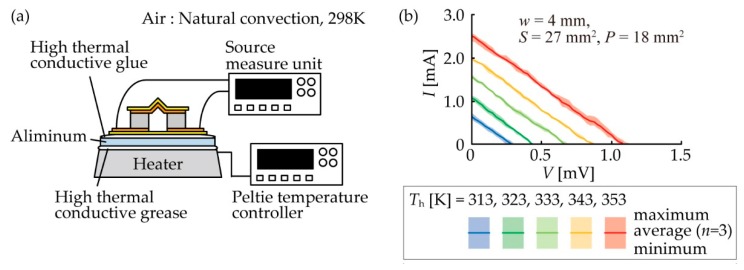
(**a**) Experimental setup for measuring the open-circuit voltage (*V*_TEG_) of TEGs. (**b**) The current–voltage (*I–V*) characteristics of the TEG with *w* of 4 mm and *S* of 18 mm^2^, which is the device shown in Figure 3ii, as an example of the measured *I–V*.

**Figure 5 micromachines-11-00263-f005:**
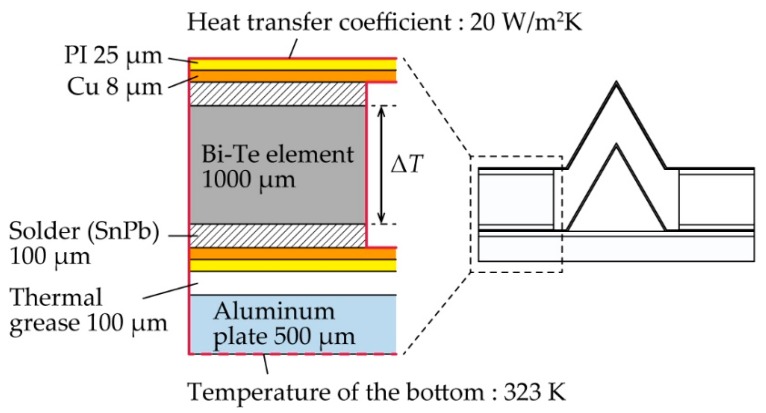
Configuration of 2D simulation models.

**Figure 6 micromachines-11-00263-f006:**
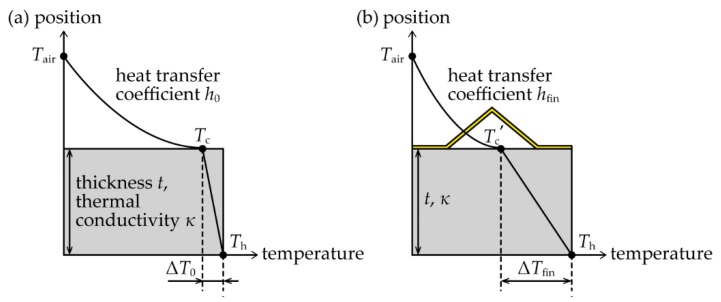
Simplified models for the derivation of *h*_fin_. (**a**) A model of a TE element without a fin. (**b**) A model of a TE element with a fin.

**Figure 7 micromachines-11-00263-f007:**
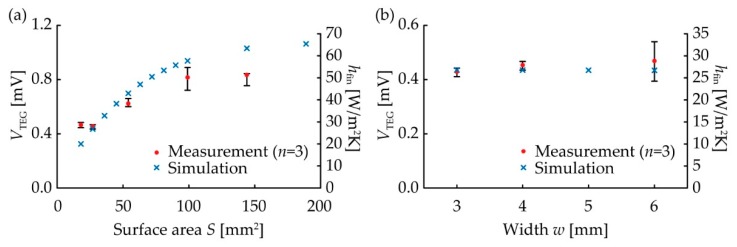
Measurement and simulation results. The temperature of the heater was 323 K and the air was 298 K. (**a**) *V*_TEG_ and *h*_fin_ vs. *S*. (**b**) *V*_TEG_ and *h*_fin_ vs. *w*.

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
