# Peer review of "An Origami Heat Radiation Fin for Use in a Stretchable Thermoelectric Generator"

_micromachines, 2020, doi:10.3390/mi11030263_

Round 1

Reviewer 1 Report

The literature review is missing. References 3-17 are not commented in the text. It has to be supplemented.
Fig. 1 and 2 are in (1) chapter, but they are described in (2) chapter.

The paper is written in the first person, singular form.

57) "polyimide: 25 μm, copper: 8 μm", thick, whide, long?
61) please, define what is "π-type structure"
63) "As shown in the top of Figure 3, we defined(...)", it is not visible; please define it clearly
70) "projected areas" - what do you mean?
89) "The measured I-V characteristic", please, show some example characteristics; describe them.
96) "heat transfer coefficient at 20 W/m2K", why 20 W/m2K?
118) "is reported to be on 118 the order of", please, cite the reference.
130) "this tendency increased as S increased", it is not clearly visible in Fig. 7
130-132) Please, explain it.
139) "the simulation values showed that width did not affect VTEG", please unwind this. It is not clear.

Reviewer 2 Report

The work presented in this proposed article concerns the construction and evaluation of a "stretchable thermoelectric generator" based on an original structure inspired by origami.

The presentation is clear. One will regret the lack of data on the electrical power potentially produced by this device, since only voltages are discussed.

The thermal approach (including the discussion on the fin effect) is relatively simplistic but gives at least orders of magnitude. It would however deserve to be deepened.

Round 2

Reviewer 1 Report

Most of the recommendations have been implemented.

The description of the literature related to the subject is very brief and could be extended.